# Evaluation by a Machine Learning System of Two Preparations for Small Bowel Capsule Endoscopy: The BUBS (Burst Unpleasant Bubbles with Simethicone) Study

**DOI:** 10.3390/jcm11102822

**Published:** 2022-05-17

**Authors:** Charles Houdeville, Romain Leenhardt, Marc Souchaud, Guillaume Velut, Nicolas Carbonell, Isabelle Nion-Larmurier, Alexandre Nuzzo, Aymeric Histace, Philippe Marteau, Xavier Dray

**Affiliations:** 1Sorbonne University, Center for Digestive Endoscopy, Saint-Antoine Hospital, APHP, 75012 Paris, France; charles.houdeville@gmail.com (C.H.); romain.leenhardt@aphp.fr (R.L.); guillaume.velut@aphp.fr (G.V.); 2Équipes Traitement de l’Information et Systèmes, ETIS UMR 8051, CY Paris Cergy University, ENSEA, CNRS, 95000 Cergy, France; marc.souchaud@ensea.fr (M.S.); aymeric.histace@ensea.fr (A.H.); 3Sorbonne University, Hepatology Department, Saint Antoine Hospital, APHP, Reference Center for Inflammatory Biliary Diseases and Autoimmune Hepatitis, 75012 Paris, France; nicolas.carbonell@aphp.fr; 4Sorbonne University, Hepatology, Gastroenterology and Saint Antoine IBD NeTwork, Saint-Antoine Hospital, APHP, 75012 Paris, France; isabelle.nion-larmurier@aphp.fr; 5Department of Gastroenterology, IBD and Intestinal Failure, Intestinal Stroke Center, Structure d’Urgences Vasculaires Intestinales (SURVI), Beaujon Hospital, APHP, 92110 Clichy, France; alexandre.nuzzo@aphp.fr; 6Sorbonne University, Department of Hepato-Gastroenterology, Tenon Hospital, APHP, 75020 Paris, France; philippe.marteau@aphp.fr

**Keywords:** small bowel, capsule endoscopy, artificial intelligence, machine learning, anti-bubble agent, simethicone

## Abstract

Background: Bubbles often mask the mucosa during capsule endoscopy (CE). Clinical scores assessing the cleanliness and the amount of bubbles in the small bowel (SB) are poorly reproducible unlike machine learning (ML) solutions. We aimed to measure the amount of bubbles with ML algorithms in SB CE recordings, and compare two polyethylene glycol (PEG)-based preparations, with and without simethicone, in patients with obscure gastro-intestinal bleeding (OGIB). Patients & Methods: All consecutive outpatients with OGIB from a tertiary care center received a PEG-based preparation, without or with simethicone, in two different periods. The primary outcome was a difference in the proportions (%) of frames with abundant bubbles (>10%) along the full-length video sequences between the two periods. SB CE recordings were analyzed by a validated computed algorithm based on a grey-level of co-occurrence matrix (GLCM), to assess the abundance of bubbles in each frame. Results: In total, 105 third generation SB CE recordings were analyzed (48 without simethicone and 57 with simethicone-added preparations). A significant association was shown between the use of a simethicone-added preparation and a lower abundance of bubbles along the SB (*p* = 0.04). A significantly lower proportion of “abundant in bubbles” frames was observed in the fourth quartile (30.5% vs. 20.6%, *p* = 0.02). There was no significant impact of the use of simethicone in terms of diagnostic yield, SB transit time and completion rate. Conclusion: An accurate and reproducible computed algorithm demonstrated significant decrease in the abundance of bubbles along SB CE recordings, with a marked effect in the last quartile, in patients for whom simethicone had been added in PEG-based preparations, compared to those without simethicone.

## 1. Introduction

Capsule endoscopy (CE) has had a major influence on small bowel (SB) exploration since its inception in 2000. The diagnostic yield (DY) of SB CE is approximately 40% in the case of obscure gastrointestinal bleeding (OGIB) [1]. The mucosa can be masked by bubbles, chyme, food residues, debris and bile, and their presence may thus decrease the DY. These factors have different causes and different protocols try to improve them. There are international guidelines on SB CE preparation, but the ideal preparation is not yet established (liquid diet, low residue diet, polyethylene glycol (PEG)-based purge, use of prokinetics, use of anti-foaming agents and timing) [2,3,4,5,6,7].

Up to now, the evaluation of these preparation modalities has been based on the analysis of thousands of images by CE expert readers, with unsatisfactory agreement between them. Indeed, a recent study showed that cleanliness scores have only moderate inter- and intra-observer reproducibility [8].

Artificial intelligence and particular machine learning (ML) has become a source of focus in the field of gastrointestinal endoscopy over the past few years [9]. Significant progress has been made for instance in the detection and characterization of lesions. Similarly, ML has achieved satisfactory results in terms of assessment of SB cleanliness, with perfect reproducibility [10].

Bubbles are one of the factors which impair mucosal visualization and thus the quality and reliability of SB CE. They result from the combination of mucus, gas and bile. Anti-foaming agents are tested for optimizing SB CE preparation. Simethicone, a blend of dimethicone and silica (a hydrophobic product), is the most widely used in this attempt. It specifically targets bubbles by decreasing the adhesion force of bubbles and promoting their bursting [11]. It is inexpensive, unabsorbed and has no toxicity or drug interactions when ingested and no serious complications have been reported with its oral administration [12].

Using ML methods, we aimed to compare the amount of bubbles in SB CE recordings, according to PEG-based preparations with and without simethicone, in outpatients with OGIB. Additionally, as bubbles are only one of the factors which impact mucosal visualization, we also measured the SB cleanliness with the two preparations.

## 2. Patients and Methods 

This was a monocentric, retrospective, non-interventional study on de-identified images.

### 2.1. Inclusion and Exclusion Criteria

Inclusion criteria were the following: outpatients with SB CE for OGIB, i.e., iron-deficiency anemia or overt bleeding, without any relevant diagnostic at upper gastrointestinal endoscopy and colonoscopy. Only outpatients were selected because hospitalization is a risk factor for inadequate preparation and incomplete SB examination [13,14].

Exclusion criteria were the following: inpatients, age below 18 years, any contraindication to CE, active bleeding (with SB flooding), technical defect (at least one missing image during recording), endoscopic CE delivery, recent SB CE investigation (in the last six months) and gastric retention (no SB frame captured).

### 2.2. Study Periods and Preparation Modalities

Patients were split into two groups, depending on their capsule PEG-based preparation, without or with simethicone (Table 1).

Group SMT− encompassed all consecutive patients fulfilling the inclusion criteria between 1 November 2018 and 31 October 2019. In this study period, they received a 500 mL preparation of PEG and ascorbate without simethicone (MOVIPREP^®^, Norgine, Harefield, United Kingdom) per oral route 30 min after ingestion of the CE [15,16].

In November 2019, the standard preparation modality for SB CE was modified in our practice. A simethicone-containing product was introduced. To avoid any overlap between the two study periods, a one-month washout period was decided: patients who had a CE in November 2019 were not eligible for the study.

Group SMT+ was composed of patients who received the above-mentioned SMT+ preparation, from 1 December 2019 to 30 November 2020: they were administered a 500 mL preparation of PEG with citric acid and simethicone (XIMEPEG^®^, Alfasigma, Bologna, Italy), per oral route, 30 min after ingestion of the CE.

### 2.3. Capsule Recording and Interpretation

All procedures were performed with SB3 Pillcam^®^ (Medtronic, Minneapolis, MN, USA). Using the Rapid^®^ software v9 (Medtronic, Minneapolis, MN, USA), SB CE recordings were pseudonymized, then read. Readings were performed at a maximum speed of 10 images per second, as recommended by the European Society of Gastrointestinal Endoscopy (ESGE) [16]. Videos were interpreted by a gastroenterology fellow (CH) and an expert reader (with more than 200 SB CE readings: NC, GV, RL, INL and XD) [17]. An international standardized nomenclature of findings was used for reporting [18]. The pertinency of lesions was classified as proposed by Saurin JC et al. for OGIB (low relevance P0, intermediate relevance P1, highly relevant P2) [19], and the guide proposed by Leenhardt R et al. [20].

### 2.4. Measurements

Clinical data collected included age, gender, type of OGIB (occult vs. overt), lower hemoglobin level, use of anticoagulants/antiaggregant, diabetes and body mass index.

A computed algorithm based on a grey-level of co-occurrence matrix (GLCM) detector strategy was used to assess the abundance of bubbles on full-length SB CE recordings (Augmented Endoscopy, Paris, France). Briefly, the algorithm is based on the textural features analyses. The matrix is built by counting the co-occurrence of two pixels of the same value appearing at the same distance and in the same direction. From this matrix, the contrast parameter is calculated to decide whether bubbles were present or not. This GLCM detector proved to be a powerful tool to differentiate still frames with a low amount of bubbles (covering less than 10% of the surface of the endoscopic image) from those with abundant bubbles (covering 10% or more of the endoscopic image surface) [21].

The ratio of the red over green pixels (R/G ratio, also known as the “SB-CAC score”) was used to assess the cleanliness of the SB in CE still frames (Augmented Endoscopy). Briefly, the color depth in the red (R) and green (G) channels of each frame was used to calculate the R/G ratio considering that a well-prepared frame has high values of red intensity (mucosal visualization) and low values of green intensity (faeces and debris) leading to a higher ratio. A R/G score higher than 1.6 was demonstrated to have the highest sensitivity and specificity to discriminate “adequately cleansed” from “inadequately cleansed” SB CE still frames in a study by Abou Ali E et al. [22,23].

At the video level, SB CE recordings were analyzed by the computed algorithm-based GLCM, for assessing the abundance of bubbles in each individual still frame. Still frames presenting with a surface covered by less than of 10% of bubbles were classified as “scarce in bubbles” and still frames presenting with a surface covered by 10% or more of bubbles were classified as “abundant in bubbles”. The proportion of “adequately cleansed” still frames with R/G > 1.6 was also calculated in all SB video sequences.

### 2.5. Study Outcomes

The primary outcome was the proportion of still frames with “abundant” bubbles in the full-length SB video sequence, as assessed by the GLCM algorithm.

Some secondary outcomes were also computed: the proportion (%) of still frames with “abundant” bubbles in each quartile (Q1 to Q4) of the SB video sequence, and the proportion of “adequately cleansed” still frames (with R/G > 1.6) in the full-length SB video sequence and in each quartile.

Other secondary outcomes were those assessed by capsule readers: the DY (defined as the proportion of video recordings with at least one P1 or P2 finding), the gastric and SB transit times, the number of images per video and the completion rate (defined as the proportion of videos with at least one colonic or stoma or anal frame).

### 2.6. Statistical Analysis

Descriptive statistics were used to indicate the patient’s demographic features and endoscopic findings. Results were expressed as means ± standard deviation (SD) for continuous variables and as percentages for categorical variables.

We used a two-way mixed analysis of variance (ANOVA) with repeated measures to compare the amount of bubbles (“abundant” vs. “scarce” in bubbles videos) and the SB cleanliness (“adequate” vs. “inadequate” based on the R/G ratio) between the two study groups. *t*-tests and Chi-square tests were otherwise used to compare continuous and categorical data between the groups, respectively.

### 2.7. Ethics

Institutional Review Board approval was obtained (Sorbonne University, n°20211116104858 12 March 2021) for this retrospective, non-interventional study on pseudonymized data.

## 3. Results

Three hundred and forty-five patients with SB CE were screened over the study period. Eighteen patients were excluded because they had their SB examination during the wash-out period. Two hundred and sixty-six patients had a SB examination for obscure gastro-intestinal bleeding (OGIB). One hundred and twenty-two inpatients were unselected. Eventually, 66 patients were considered in the SMT− group and 78 in the SMT+ group. Thirty-nine additional patients met exclusion criteria. Overall, 48 patients were included for analysis in the SMT− group and 57 in the SMT+ group (see flowchart, Figure 1).

There were 51 male and 53 female patients in the study population, with a mean age of 62.4 ± 15.3 years. SB CE was indicated for overt OGIB in 31 patients (29.5%). Antiaggregant and anticoagulant treatments were used in 33 (31.4%) and 20 (19.0%) patients, respectively. There were no statistically significant differences in demographics and in clinical data between the two study groups (Table 2).

### 3.1. Capsule Outcomes

The mean gastric transit time was around 40 min in both groups (*p* = 0.47). The SB transit time and the number of SB images per video were higher in the SMT+ group, compared to the SMT− group, although these differences were not statistically significant. The caecum was reached in 87.5% of patients in the SMT− group, and in 92.9% of patients in the SMT+ group (*p* = 0.51). No significant differences were observed in the DY (P1 or P2 lesions) between SMT− and SMT+ groups (41.7% vs. 42.1%, *p* = 0.96) (Table 3).

### 3.2. Abundance of Bubbles

The proportion of still frames with “abundant” bubbles in the SB, as assessed by the GLCM algorithm was significantly lower in the SMT+ group as compared to the SMT− group (ANOVA, *p* = 0.04) (Figure 2). This difference between the two groups was marked in the third quartile (Q3) (22.1% vs. 27.7%, *p* = 0.12), and even more accentuated in the last quartile (Q4) (20.6% vs. 30.5%, *p* = 0.02).

### 3.3. R/G Ratio 

Although the percentage of images with a red over green (R/G) ratio over 1.6 (“adequately cleansed” still frames) significantly decreased over time (*p* = 0.04), no significant difference was found between the two study groups (ANOVA, *p* = 0.26) (Figure 3).

## 4. Discussion

The best modalities for preparing patients for SB CE still need to be established, and particularly regarding the use of simethicone which aims at reducing bubble formation and persistence. Improving SB cleanliness is not only based on bubble reduction, but also on increasing brightness, decreasing the amount of residues, fluids and bile [24]. Each of these factors can be approached specifically. In this study, we addressed the question of bubbles reduction as the primary endpoint and then assessed whether this reduction had an impact at a larger scale (cleanliness and DY, as secondary endpoints).

To the best of our knowledge, this is the first study where a reliable artificial intelligence using a ML system demonstrates that a simethicone-based preparation significantly reduces the abundance of bubbles over the entire SB, and particularly in the last quartile.

The main strength of this study is that it uses two algorithms that have been shown to be highly sensitive, highly specific and perfectly reproducible [10,21,22,23]. These algorithms are comprehensive, given that they analyze each individual frame for specific patterns (i.e., bubbles and R/G ratio), thus outperforming any human ability for such tedious tasks. Indeed, it has been demonstrated that the human evaluation of SB CE cleanliness is poorly reproducible [25].

Some weaknesses must be acknowledged. First, it is a retrospective monocentric study, but it covers a long period (two years) of routine use of SB CE. The number of analyzed SB CE videos (from 105 patients) was limited, and this limits the statistical power, not only for the primary (the amount of bubbles) but also for the secondary endpoints (most importantly SB cleanliness and DY). However, our results bring preliminary data based on AI that will allow calculations of sample size and power for further prospective studies. Second, the study focuses on selected patients (outpatients with OGIB), which questions the generalizability of our results. However, this selection of patients allowed us to avoid bias due to lower cleansing rates in inpatients [8]. Third and foremost, the two preparation modalities used, did not have exactly the same characteristics (Table 1). However, they are relatively similar apart from their content in ascorbic acid (MOVIPREP^®^) and in citric acid (XIMEPEG^®^), which have similar weak osmotic laxative effects. The only additional and active substance in XIMEPEG^®^ is simethicone.

Our results compare well to what is known in the literature. The cleanliness of the distal part of the ileum is poorer than the SB segments upstream [26,27,28]. Our AI-based analysis shows a significant decrease in the proportions of supposedly clean images (i.e., with a R/G ratio above 1.6) along the SB quartiles in both groups (Figure 3). However, this reduction regarding cleanliness (as measured by the R/G ratio) was similar between groups. This may be due to a lack of power related to the small sample size in our study. Still, up to now, studies based on human analysis remain controversial regarding any SB cleanliness improvement with simethicone [5,7,29]. In a recent randomized trial, a preparation with a high dose of simethicone did not significantly improve the SB cleanliness compared to a standard preparation, as assessed by human readers [30]. Our study, based on AI analysis, strengthens this finding: the overall cleanliness of the SB (as measured by the R/G score) was not significantly improved when using the simethicone-based preparation. In addition, as shown by the systematic review by Wu L et al. [31], we found that simethicone did not improve the completion rate. Lastly, our findings regarding the DY are in line with those of Song H et al. where no improvement is found when adding simethicone [32]. It may be due, however, to the limited sample size of our study.

We believe that randomized control studies with simethicone and placebo, using computer algorithms (rather than human readings) for bubble assessment, should be considered in patients with OGIB, as well as in patients with Crohn’s disease, where the evaluation of the terminal ileum by SB CE is challenging and is of the utmost importance [33,34]. Further studies may be conducted to improve the level of evidence in this area.

Overall, our findings, based on highly accurate and reproducible computed algorithms, show that simethicone has a marked effect on bubble reduction and particularly in the challenging distal ileum, when reading a CE recording. This antifoaming agent is inexpensive and extremely safe (no absorption, no toxicity, no drug interactions) [12] We therefore believe that the balance between benefits and risks/costs of using simethicone for SB preparation is favorable. This is in line with the recommendations of the ESGE and the American Gastroenterological Association (AGA) to administrate an antifoaming agent for CE [16,35]. However, in our study, simethicone only had a modest effect and no direct impact on SB cleanliness and DY.

## Figures and Tables

**Figure 1 jcm-11-02822-f001:**
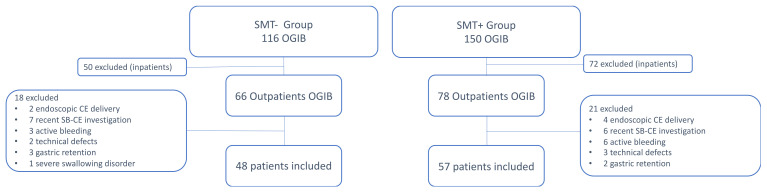
Study flow chart. SMT− stands for “without simethicone” whereas SMT+ means “with simethicone”. CE: capsule endoscopy; OGIB: obscure gastrointestinal bleeding; SB: small bowel.

**Figure 2 jcm-11-02822-f002:**
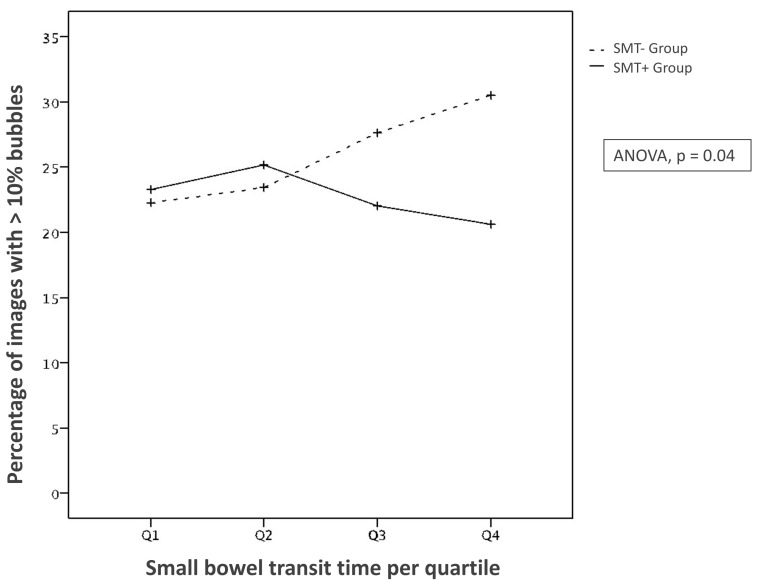
Interaction plot between the two groups about the percentage of images with more than 10% of bubbles over transit time (by quartile). SMT− stands for “without simethicone” whereas SMT+ means “with simethicone”.

**Figure 3 jcm-11-02822-f003:**
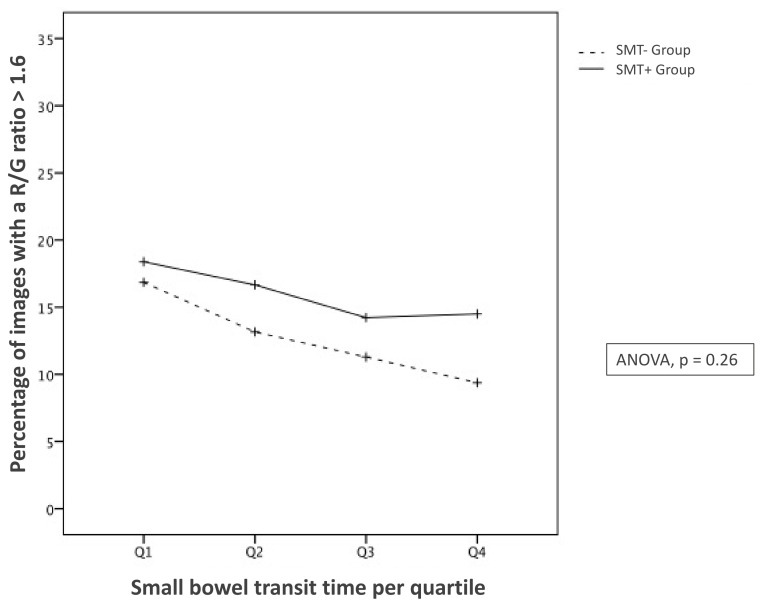
Interaction plot between the two groups about the percentage of images with a R/G ratio over 1.6 by quartile. SMT− stands for “without simethicone” whereas SMT+ means “with simethicone”.

**Table 1 jcm-11-02822-t001:** Study periods and composition for 500 mL of Ximepeg^®^ and Moviprep^®^ solutions. Active substances are in grammes, and electrolytes are in Millimoles per liter. SMT− stands for “without simethicone” whereas SMT+ means “with simethicone”.

	SMT− Group	Wash Out Period	SMT+ Group
	1 November 2018 to 31 October 2019	November 2019	1 December 2019 to 30 November 2020
	MOVIPREP^®^ (500 mL)	Not Included	XIMEPEG^®^ (500 mL)
Active Substance (g)			
Macrogol	50		26.3
Sodium sulfate	3.75		1.88
Sodium chloride	1.34		0.37
Potassium chloride	0.5		0.19
Ascorbic acid	2.35		0
Sodium ascorbate	0.93		0
Sodium citrate	0		0.93
Citric acid	0		0.41
Simethicone	0		0.04
Electrolyte (mmol/L)			
Sodium	90.5		84
Sulfate	26.4		26
Chloride	29.9		17
Potassium	7.1		5.6
Ascorbate	14.9		0
Citrate	0		10.6

**Table 2 jcm-11-02822-t002:** Clinical data. SMT– stands for “without simethicone” whereas SMT+ means “with simethicone”. OGIB: obscure gastro-intestinal bleeding; CE: capsule endoscopy.

	SMT− Group (*n* = 48)	SMT+ Group (*n* = 57)	*p*
Age (years)	63.2 ± 13.3	61.7 ± 16.8	0.61
Men (*n*)	19 (39.6%)	32 (54.3%)	0.12
Women (*n*)	29 (60.4%)	25 (43.9%)	0.12
Diabetes (*n*)	9 (18.8%)	17 (29.8%)	0.25
Body mass index (kg/m^2^)	25.5 ± 4.6	27.1 ± 5.7	0.12
Anticoagulant (*n*)	8 (16.7%)	12 (21.1%)	0.62
Antiaggregant (*n*)	14 (29.2%)	19 (33.3%)	0.68
Lower hemoglobin rate (g/dL)	9.0 ± 2.5	9.4 ± 2.7	0.43
Overt OGIB (*n*)	11 (22.9%)	20 (35.1%)	0.27
If overt OGIB, average time in before CE (days)	18 ± 12	42 ± 31.1	0.10

**Table 3 jcm-11-02822-t003:** Number of images per video (mean and SD), mean transit times per segment, diagnosis yield (%) and completion rate (%) in each group. SMT− stands for “without simethicone” whereas SMT+ means “with simethicone”.

	SMT− Group (*n* = 48)	SMT+ Group (*n* = 57)	*p*
Diagnostic yield			
P1 lesion rate (%)	14.6	15.8	0.86
P2 lesion rate (%)	27.1	26.3	0.92
P1 or P2 lesion rate (%)	41.7	42.1	0.96
Capsule progression			
Gastric transit time (min)	40 ± 60	39 ± 46	0.47
Small bowel transit time (min)	199 ± 116	232 ± 123	0.08
Completion rate (%)	87.5	92.9	0.51
Number of images per videos	5887 ± 3405	6527 ± 3801	0.18

## Data Availability

Not applicable.

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
