# Peer review of "Evaluation by a Machine Learning System of Two Preparations for Small Bowel Capsule Endoscopy: The BUBS (Burst Unpleasant Bubbles with Simethicone) Study"

_jcm, 2022, doi:10.3390/jcm11102822_

Round 1

Reviewer 1 Report

Report of a novel approach to image analysis in endoscopy. Practical results are fairly well known, but the technology is new.

Minor comments:

Please spell out 'PEG' at first occurence in the text

'AGA' is missing in the list of abbreviations

Author Response

1- Report of a novel approach to image analysis in endoscopy. Practical results are fairly well known, but the technology is new.

Answer: Thank you.

2-      Minor comments: Please spell out 'PEG' at first occurrence in the text and 'AGA' is missing in the list of abbreviations

Answer : We apologize for these oversights. Modifications have been done.

Reviewer 2 Report

In their study, Hou et al. evaluate two small intestinal capsule endoscopic formulations using a machine learning system. The results showed a significant reduction in the number of bubbles along the SB-CE recordings in patients with PEG-based formulations to which simethicone was added, compared to patients without simethicone. The paper is concise and clearly written throughout. My comments on the paper as a whole are as follows.

1) The SMT+ group has a slightly longer small intestinal transit time, is there a reason for this?

2) It would be better to include the SMT abbreviation in the figure or table legend.

3) The article by Sey M et al (PMID: 33793636) that discusses the optimal dosage of cimethicone prior to capsule endoscopy should also be mentioned in the discussion.

Author Response

1-       In their study, Houdeville et al. evaluate two small intestinal capsule endoscopic formulations using a machine learning system. The results showed a significant reduction in the number of bubbles along the SB-CE recordings in patients with PEG-based formulations to which simethicone was added, compared to patients without simethicone. The paper is concise and clearly written throughout. My comments on the paper as a whole are as follows.

Answer: The authors thank the reviewer for the kind comment and his (her) interest in our work.

2-   The SMT+ group has a slightly longer small intestinal transit time, is there a reason for this?

Answer: We agree with the reviewer’s comment :  in the literature, effect of simethicone on small bowel transit time is controversial. In some studies it seems to increase transit time whereas in others it doesn’t have an effect. It might be because  transit time is not the primary criteria of these non-prospective studies.  

  1. It would be better to include the SMT abbreviation in the figure or table legend.

Answer: We agree with you, and we have added the following sentence the figures and tables :” SMT– stands for without simethicone whereas SMT+ means with simethicone. 

  1. The article by Sey M et al (PMID: 33793636) that discusses the optimal dosage of simethicone prior to capsule endoscopy should also be mentioned in the discussion.

Answer: We have double-checked all references and the discussion where the following sentences are already : In a recent randomized trial, a preparation with a high dose of simethicone did not significantly improve the SB cleanliness compared to a standard preparation, as assessed by human readers [30: reference to Sey M and al]. Our study, based on AI-analysis, strengthens this finding: the overall cleanliness of the SB (as measured by the R/G score) was not significantly improved when using the simethicone-based preparation”.

Round 2

Reviewer 2 Report

The authors responded appropriately to the reviewers' comments.

Round 1

Reviewer 1 Report

Dear Authors

I appreciated your work which seemed to me to be well elaborated and with
clinical significance.

however, the sample number  limits the conclusions.
It would not be expected that the reduction of artifacts (bubbles) would
not have an impact on mucosal observation and improvement in diagnostic acuity

The references are adequate but I suggest some corrections, because the final page is missing in some

Many regards

Reviewer 2 Report

I have carefully read a paper by Houdeville et al. concerning the evaluation of two preparations for small bowel capsule endoscopy by a machine learning system. Several reasons disqualify the manuscript from being published in the Journal of Clinical Medicine: i.e., the unclear aim of the study. There are no widely accepted guidelines for small bowel preparation for capsule endoscopy. I don't see an explanation why one should use Machine Learning System to evaluate just one of the preparation measurements (bubbles), entirely omitting other (i.e., chyme, food residues, debris, and bile pigments). Moreover, it is not clear why the authors decided to support the idea of using simethicone, as no higher diagnostic performances were observed in this study. 

Round 2

Reviewer 2 Report

Answers and adjustments made by the Authors did not change overall thurst of the paper and its conclusions.